# Urea Nitrogen Metabolite Can Contribute to Implementing the Ideal Protein Concept in Monogastric Animals

**DOI:** 10.3390/ani12182344

**Published:** 2022-09-08

**Authors:** Pablo Jesús Marín-García, Lola Llobat, Mari Carmen López-Lujan, María Cambra-López, Enrique Blas, Juan José Pascual

**Affiliations:** 1Department of Animal Production and Health, Veterinary Public Health and Food Science and Technology (PASAPTA), Facultad de Veterinaria, Universidad Cardenal Herrera-CEU, CEU Universities, 46113 Valencia, Spain; 2Institute for Animal Science and Technology, Universitat Politècnica de València, Camino de Vera s/n, 46022 Valencia, Spain

**Keywords:** monogastric, amino acid, ideal protein, broiler, pig, rabbit

## Abstract

**Simple Summary:**

Can urea nitrogen metabolite contribute to implementing the ideal protein concept in monogastric animals? This work aims to critically analyse how this metabolite can contribute to accurately implementing the ideal protein concept in monogastric animals, particularly in pig, poultry, and rabbit nutrition. This information will contribute to evaluating its potential and limitations as biomarker, as well as to standardizing the use of this metabolite in precise amino acidic monogastric nutrition.

**Abstract:**

The ideal protein concept refers to dietary protein with an amino acid profile that exactly meets an animal’s requirement. Low-quality protein levels in the diet have negative implications for productive and reproductive traits, and a protein oversupply is energetically costly and leads to an excessive N excretion, with potentially negative environmental impact. Urea Nitrogen (UN), which corresponds to the amount of nitrogen in the form of urea circulating in the bloodstream, is a metabolite that has been widely used to detect amino acid imbalances and deficiencies and protein requirements. This review aims to critically analyse how UN can contribute to accurately implementing the ideal protein concept in monogastric animals, particularly in pig, poultry, and rabbit nutrition (14,000 animals from 76 published trials). About 59, 37, and 4% of trials have been conducted in pigs, poultry, and rabbits, respectively. UN level was negatively correlated to main performance traits (Pearson Correlation Coefficient [PCC] of −0.98 and −0.76, for average daily gain and feed conversion ratio, respectively), and lower UN level was related to higher milk yield and concentration. High level of UN was positively correlated to N excretion (PCC = 0.99) and negatively correlated to protein retention (PCC = −0.99). Therefore, UN in blood seems to be a proper indicator of amino acid imbalance in monogastric animals. Great variability in the use of UN was observed in the literature, including uses as determination medium (blood, plasma, or serum), units, and feeding system used (ad libitum or restricted), among others. A standardization of the methods in each of the species, with the aim to harmonize comparison among works, is suggested. After review, UN measurement in plasma and, whenever possible, the utilization of the same nutritional methodology (ad libitum conditions or restriction with blood sampling after refeeding at standardised time) are recommended. More studies are necessary to know the potential of UN and other bioindicators for amino acid deficiencies evaluation to get closer to the ideal protein concept.

## 1. Introduction

The ideal protein concept refers to dietary protein with an amino acid profile that exactly meets an animal’s requirement [1]. When applying this concept, all essential and non-essential dietary amino acids can be equally co-limiting. Formulating according to the ideal protein concept can improve animal performance, increasing protein retention [2], and can contribute to reducing N excretion [3,4,5]. Nevertheless, to use the ideal protein concept, it is necessary to combine precise knowledge on feed evaluation and an animal’s nutritional requirements. An animal’s amino acids requirements are dynamic, since they can vary according to different factors including genetics, age, and physiological state [6]. Amino acid requirements have also shown to vary individually [7] within the same animal type in growing pigs. Currently, it is well accepted that amino acid requirements and feed evaluation should be determined at ileal true digestible level, because this information provides a more accurate estimation compared to total amino acids content at dietary or faecal level.

In addition, there is an increasing interest in the use of low-protein diets as a tool to improve the sustainability of animal production [8]. Previously, protein requirements were determined by dose–response experiments [9,10,11,12,13,14,15,16,17,18]. Commonly, dose–response studies only deal with a specific amino acid, without considering amino acid interactions [19]. These limitations could be overcome by acquiring information on the metabolic profile of animals that are fed with different levels of essential amino acids. This approach could reveal metabolic phenotypes related to the presence of specific limiting amino acids. At present, metabolomics can be used as a tool to investigate the relationship between nutritional status and metabolic phenotyping [20]. Among these metabolites, urea nitrogen (UN) level in blood has been widely used to detect amino acid imbalances and deficiencies.

This review aims to critically analyse how UN determination in blood can contribute to accurately implementing the ideal protein concept in monogastric animals, particularly in pig, poultry, and rabbit nutrition. This information will contribute to evaluating its potential and limitations, as well as to standardizing the use of this metabolite in precise amino acidic monogastric nutrition.

To achieve this goal, a critical review was carried out using all the articles published on monogastric species where UN was analysed in any of its forms. The different aspects studied in these works were studied and analysed jointly through correlation coefficients.

## 2. Monogastrics’ Protein Metabolism

Protein are macro-biomolecules, and monogastrics carry out protein synthesis through translation, creating their own protein using the amino acids available after cellular digestion. Amino acids become available as end-products after protein digestion in an animal’s digestive tract. Protein and amino acids oversee many functions in animals (structural, regulatory, transporter, defensive, enzymatic, or even contractile). There are two types of amino acids: non-essential and essential [21,22] Essential amino acids depend on the animal species [10,11,23,24]. Starting from the diet, animals can synthesize their own protein. However, protein efficiency, usually measured by dividing protein retention by protein intake and multiplying by 100, is low, and only a part of ingested protein can be digested, metabolised, and effectively used by the animal [2,25,26,27]. 

Figure 1 shows how only a portion of total ingested crude protein (CP) is digested and converted into amino acids that can be absorbed in the small intestine. The amino acids that are absorbed until ileum correspond to the ileal digestible CP. The rest (ileal indigestible CP) undergo processes into the large intestine where they can be used or modified by microbial action and finally eliminated in faeces (faecal indigestible CP). It is therefore reasonable to believe that ileal amino acid digestibility (and especially true ileal digestibility, which takes into account the endogenous losses) allows us to know the quantity of amino acids that are available for the animal [28], giving valuable information on both the animal’s requirements and nutritional balance of the diet.

Animals synthetize specific proteins depending on their requirements [29]. Protein synthesis mainly depends on animal requirements, determined by DNA transcription, and amino acids provided by the diet [30]. When there is no amino acid limitation, protein synthesis can happen adequately. However, if there is any amino acid limitation, the specific protein will not be synthesized.

## 3. When Protein Is Synthesized from Balanced Diets: Protein Requirements

If the amino acids of the protein to be synthesized by the animals match with the amino acids available at the end of digestion, protein synthesis will occur. Total protein requirements are calculated factorially by summation of the requirements for maintenance and production as well as the immune status and the typical characteristics of the species.

Protein requirements for maintenance are the amino acids used by the animal when its body composition remains constant. The determination of protein requirements for maintenance in monogastrics is controversial, and the results vary depending on the consulted bibliographical source [16,31,32,33,34,35,36,37,38,39,40].

The growth pattern of an animal also determines its protein requirements. The growth of all body components can be investigated by slaughtering and analysing animals at successive stages of growth, determining retained nitrogen [41]. This retained nitrogen is related to body weight and nitrogen intake by the animals [32,33,34,42,43,44,45].

An animal’s requirements are also affected by reproduction. In spite that, the quantities of protein required to produce spermatozoa by mammals are small and of little significance [30], and all female mammals increase protein requirements in the reproductive period. Females must mobilize protein for ovulation, implantation process, foetal development, and milk production. [46,47,48,49].

The health status (immunity responses) of the animals also influences their protein requirements and vice versa [25,50,51,52]. Research has shown differences in immunity responses to be functions of dietary protein level in all monogastric species [53].

There is another peculiarity in a monogastric that must be considered when calculating protein requirements, in the case of lagomorphs, which practice caecotrophy. The caecotrophy contribution to the total CP intake is around 17% [25]. Furthermore, protein of soft faeces is rich in essential amino acids [54,55].

## 4. When Protein Is Not Synthesized: Urea Nitrogen Formation

No dietary amino acids are absorbed in a total form for utilization by the animal [56]. In addition, not all of the absorbed fraction is used by the animal (e.g., due to the presence of some limiting amino acid). These amino acids (absorbed but not used) together with the ones coming from cell renewal are catabolized, in the urea cycle (Figure 1), in the mitochondria and cell cytosol of liver cells. Urea is the end-product, and it passes into the bloodstream and from there to the kidneys, which excrete it in urine. UN can be measured in blood (blood urea nitrogen; BUN), concentrated plasma (plasma urea nitrogen; PUN), or serum (serum urea nitrogen; SUN), depending on whether clotting factors are included or not [57]; in this review, UN measurement will encompass all three. The urea cycle requires the utilization of energy in urea production. Therefore, diets that induce high urea levels will result in a reduction in animal performance (related to growth traits [58], health status [25], and reproductive traits, among other factors), while an oversupply will be energetically costly and will lead to pollution [1,3,59]. That is why works that enable knowing the amino acid needs of animals are very important.

As it has been said previously, amino acid requirements are usually determined by dose–response trials [9,10,11,12,13,14,15,16,17,18]. In addition to dose–response trials, animals have showed a certain ability to modulate their amino acid profile intake when they have the opportunity to choose between diets with different levels of amino acids in a choice feeding trial [60,61,62]. This ability of animals (to choose the diet that fits their nutritional requirements) should also be considered in future works that study the ideal protein profile of the diets. Nevertheless, amino acids requirements determination is complex, because there are many interactions among them and between amino acids and other nutrients. In general terms, when comparing the animal amino acid requirements with the amino acid composition of their targeted diets, it is shown that the ‘first-limiting’ amino acid for pigs is Lysine, whereas for poultry and rabbits, it is commonly Methionine, although Lysine and Arginine may also be limitans.

As it has been explained previously, on one hand, UN is related with the parts of a protein that were absorbed but non-utilized, and on the other hand, implementing the ideal protein concept in monogastric animals’ nutrition could improve animal performance and reduce pollution. In this moment, a question arises: Could UN be used as a measurement of the ideal protein concept in monogastrics?

## 5. Use of Urea Nitrogen to Detect Amino Acid Imbalances

Animals fed with a balanced diet show low UN levels in blood, indicating a decrease in protein catabolism, more efficient total N utilization, and thus a decrease in urea synthesis. Although the first research using UN as an indicator of CP requirements was carried out in the 1970s [63,64], the use of this metabolite has become more common in recent years, because it is a rapid and cheap response criterion to determine amino acids requirements [65,66].

Table 1 summarizes studies where UN has been used as an indicator for improving productive or reproductive traits (to these works must be added those that relate UN with N excretion). In these studies, around 14,000 animals were used in 76 different trials. On average, the number of treatments used in each trial was around five, with approximately 48 animals per treatment. About 59, 37, and 4% of the trials were conducted in pigs, poultry, and rabbits, respectively (excluding experiments carried out on rats). In both pig and poultry studies, a range from reproductive animals to noticeably young individuals can be found. Nevertheless, in rabbit studies, UN has only been determined in growing rabbits. According to the dietary treatment, only 5% of the studies compared protein source, 17% studied CP, and 78% studied amino acids level.

### 5.1. Causes of Variability in UN Measurements

Despite the extensive use of this metabolite, there are discrepancies in its use that must be considered when comparing studies. Like any blood metabolite, UN presents daily circadian variations. Therefore, evaluation of UN evolution in time in each animal species is needed, considering particularly feeding behaviour. Of the total published works, only 5% have evaluated this issue [56,63,74,87]. In fact, no research was found in broilers or hens addressing this matter.

As it has been explained previously, UN level depends on protein intake, and it is for this reason that feed behaviour can alter UN measurements. For example, 20% more PUN (*p* < 0.05) has been observed in samples obtained 4 h after a refeeding in animals that were submitted to a fasting trial than those animals (for the same animal type and with the same experimental diet) that were fed ad libitum. About 30 and 70% of the studies were performed in animals fed ad libitum and restricted diets, respectively [87]. In those animals fed ad libitum, the circadian evolution of UN followed a pattern very similar to that of feed intake, when animals were fed with balanced diets. However, in the case of animals subjected to fasting, the UN pattern after refeeding increases for the first 3–4 h after feeding and thereafter reaches a plateau. UN concentrations as a function of time after refeeding must be considered to compare results between works (the sample collection presents a large variability, from 1 h to 16 h after refeeding, in the consulted literature).

Regarding the standardization of feeding management, we recommend feeding in ad libitum conditions or drawing blood samples 4 h after refeeding (since it has been shown that the highest UN levels are reached after ingestion). It would be advisable to specify the type of feed management.

In addition to the above, these studies showed different experimental designs such as randomized blocks, incomplete blocks, Latin square, repeated Latin square, and even repeated measures [88]. Another observed discrepancy may be due to the type of measurement. As previously mentioned, UN can be determined in blood, plasma, or serum. Independently of the species and physiological state, 66, 28, and 14% of the works determined UN in plasma, serum, or blood, respectively. This is a factor of variability that causes deviations in UN results, which must be considered when comparing trials. Furthermore, there is also great diversity in the units used to express UN. Most studies express it in mg/dL, although there are some others that express it with other units such as mmol/L [75]. Is possible to convert from one unit to another, with 1 mmol/L = 6.006 mg/dL.

Despite the mentioned methodological differences, of the total of these studies, 86% showed that UN could be used for the detection of amino acid imbalances (see Table 1). Next, we proceed to analyse the different relationships of this metabolite with performance and environmental nutrient load.

### 5.2. Urea Nitrogen and Performance/Reproductive Traits

Most of the studies where UN was used were carried out in growing animals. The main traits controlled to evaluate growing performance were feed intake, average daily gain (ADG), and feed conversion ratio (FCR), as well as other less frequent traits such as carcass quality. In general terms, UN values are negatively correlated with protein utilization and growth performance traits. Figure 2 (*n* = 348 animals) shows the linear relationship between UN and ADG (Figure 2A) and FCR (Figure 2B) obtained with different experimental diets varying in three essential amino acids (Lysine, Sulphur amino acids, and Threonine) in growing rabbits [2,60,87,89]. Data on this figure reflects how UN levels are negatively correlated to ADG (Pearson Correlation Coefficient [PCC] = −0.98; *p* < 0.05) and positively correlated to FCR (PCC = 0.81; *p* < 0.05). Along this same line, Figure 3 (*n* = 2154 animals) shows the linear relationship between UN and FCR obtained in different trials in growing pigs [65,66,73,75,76,78]. As it has been shown in rabbits, UN levels in pigs are positively correlated to FCR (PCC = 0.70; *p* < 0.05). It should be considered that in making comparisons between animals of different ages and genetic types, the relationships between UN and ADG or FCR may be compromised. That is why the relationship with FCR is stronger (FCR is corrected for the marked differences in ADG and feed intake in animals of different ages). As can be extracted from Figure 2 and Figure 3, for every 2.5 and 1.55 mg/dL more units of UN in plasma, the FCR increases by 0.1 in rabbits and pigs, respectively (*p* < 0.05). This could be because low UN levels imply more efficient use of protein, lower imbalance in the amino acid profile, and less energy addressed to catabolizing non-used amino acids and UN metabolization. Therefore, nutrients are more available for growth and fitted to the requirements. Other trials have been conducted to investigate the relationship between UN and reproductive performance traits in sows [44] or laying hens. No trials have been performed on reproductive rabbits until recently. In these studies, parameters such as milk yield, composition yield, and egg production have been analysed. These studies have shown that lower levels of UN are usually related to higher milk yield and concentration (even increasing the quantity of immunoglobulins in colostrum [90]).

### 5.3. Urea Nitrogen and Environmental Nutrient Load

Nitrogen pollution from livestock should be studied, as it is a serious problem for the environment [91]. Contamination of groundwater is a serious environmental issue of global concern because of its direct adverse effect on human health and biodiversity, among other effects [92]. A total of nine studies have been conducted to evaluate the relationship between UN level and N excretion. All the works observed a relationship between UN and nitrogen contamination [66,71,74,88,90,93,94,95,96]. As it can be seen in Figure 4 (starting from two experimental trials conducted in pigs [71,86]), UN level in blood seems to be positively correlated to N excretion (PCC = 0.99; *p* < 0.05) and negatively correlated to protein retention (PCC = −0.99; *p* < 0.05). For all the above, diets that induce higher UN levels have negative implications for productive and reproductive traits and lead to an excessive N excretion, with potentially negative environmental impact.

## 6. Future Perspectives

As a summary, UN level in blood has been shown to be a promising indicator of protein and amino acid use, sensible to amino acid imbalance in pigs, poultry, and rabbits. It has been validated as a useful response criterion related with performance and nitrogen excretion data. Its use would have the potential to obtain indications of possible deficiencies or imbalances of dietary amino acids, and some recommendations for extending the use of this metabolite as a rapid and economical response indicator for amino acid imbalances have been proposed.

The ideal protein concept could be used not only in animal production and environment, as it can also be applied in other areas such as animal conservation. Some wild animal populations seem to be limited by protein supply of their ecosystems. In this case, meeting nutritional requirements would improve adaptive success of wild animals, and specific vegetation management programs could be carried out for the conservation of these animals [97,98].

On the other hand, there are a multitude of metabolites that could complement UN use to improve the concept of the ideal protein. Metabolomics arises as a valuable tool to identify and quantify biomarkers related to the presence of limiting amino acid. Some studies have proposed as candidate metabolites plasma glychocolic acid, taurocholic acid, and ascorbic acid, among others [99]. However, more studies are needed to complement UN with other metabolites to implement the ideal protein concept in monogastric animals.

An important factor is the interaction between protein nutrition and heat stress caused by climate change. The activity of feeding (amino acid included) and the metabolism caused by digestion and assimilation of food increase an animal’s heat production [100]. It is known that the metabolism of amino acids in excess produces the most heat among all nutrients, negatively influencing animal performance. The ideal protein concept could reduce the heat of metabolism, allowing animals to withstand thermal stress better, and UN could be used along this line [101].

## 7. Conclusions

The use of urea nitrogen in blood is proven as an indicator of amino acid imbalance in monogastric animals. However, there are some discrepancies in the methodology used in the different studies, which make it difficult to compare them. The standardization of the methods in each of the species is recommended. From the analysis of the literature, this study recommends the measurement of UN in plasma (PUN) and whenever possible trying to always use the same feeding management (ad libitum conditions or drawing blood samples 4 h after refeeding). More studies are necessary to know the potential of UN and other bioindicators for amino acid deficiencies evaluation to get closer to the ideal protein concept.

## Figures and Tables

**Figure 1 animals-12-02344-f001:**
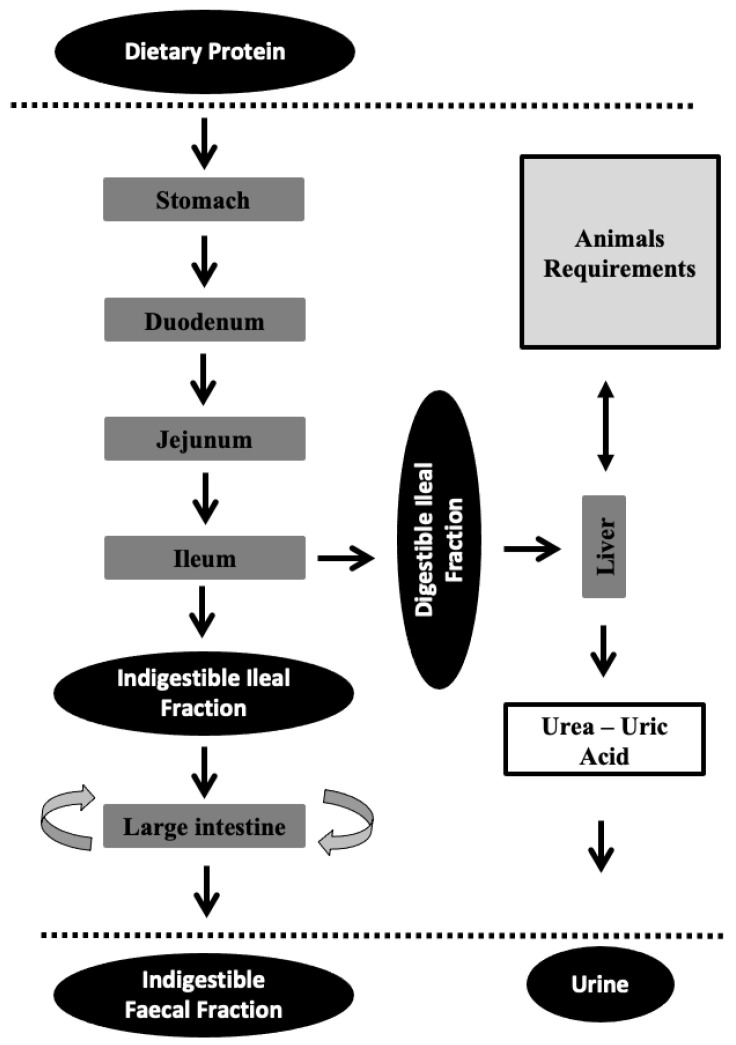
The protein partitioning and utilization in monogastric animals. The dotted lines symbolize the interior of the animal. The different arrows in the large intestine refer to the conversion of some amino acids into others produced by the microbiota.

**Figure 2 animals-12-02344-f002:**
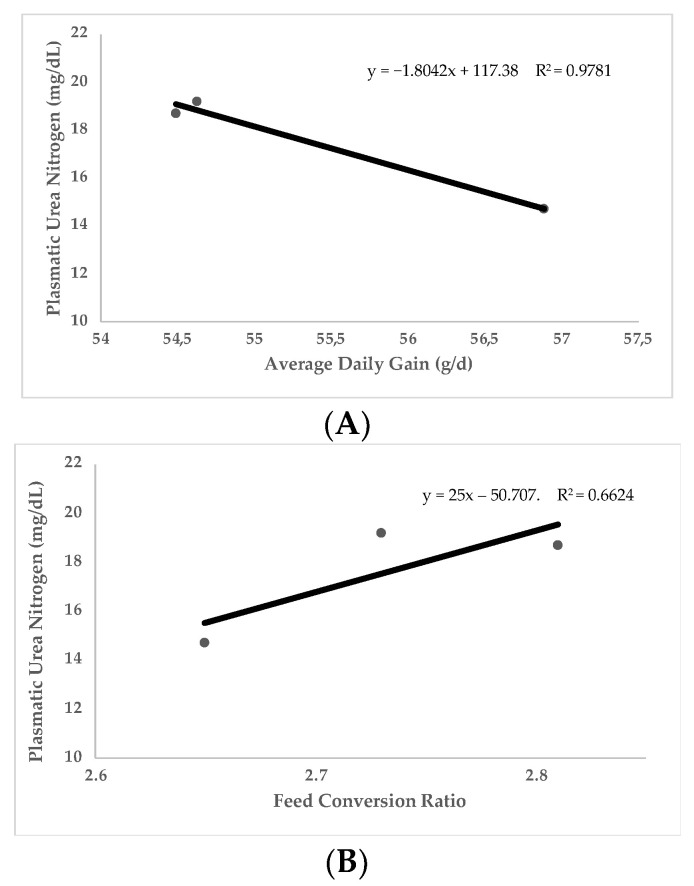
Relationship of plasmatic urea nitrogen (PUN) with average daily gain (**A**) and with feed conversion ratio (**B**). Data obtained from four published studies in growing rabbits [2,60,87,89].

**Figure 3 animals-12-02344-f003:**
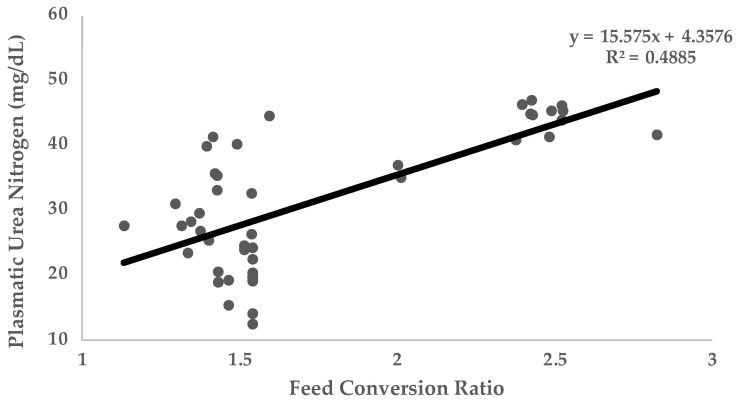
Relationship of plasmatic urea nitrogen (PUN) with feed conversion ratio. Data obtained from four published studies in growing pigs [65,66,73,75,76,78].

**Figure 4 animals-12-02344-f004:**
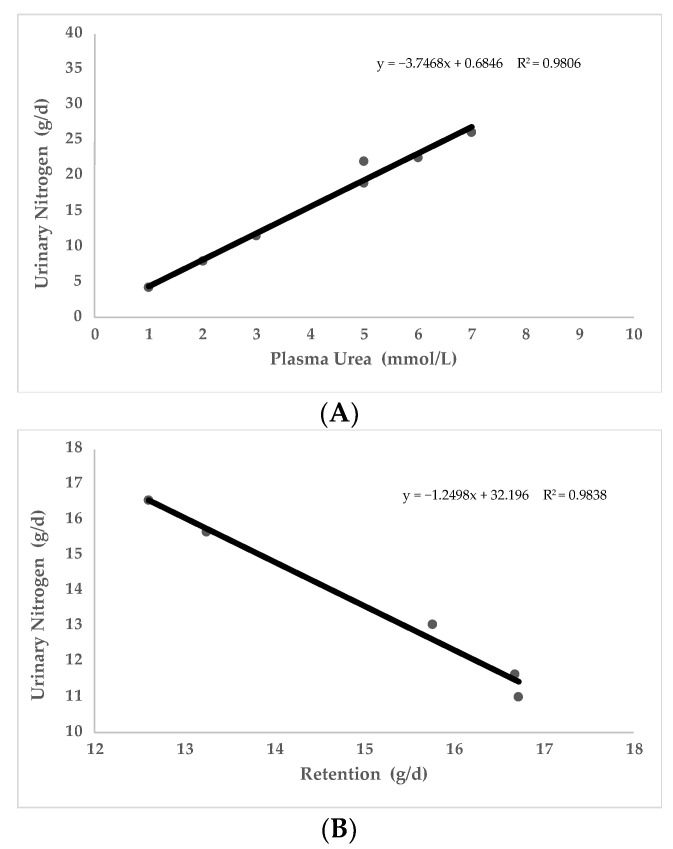
Relationship of plasma urea nitrogen (PUN) with urinary nitrogen excreted (**A**) and relationship of urinary nitrogen excreted with Nitrogen retained (**B**). Data obtained from two published studies in sows and growing pigs [71,86].

**Table 1 animals-12-02344-t001:** Parameters studied in the published works where urea nitrogen (UN) is used to determine nutritional requirements in monogastrics and related with performance and reproductive traits.

References	Trial	Anim. Type	Nº	Nutrient	Level	Treatments	UN	Feeding	XUN Units	X	SE	Correlation
[67]	1	Broiler	320	CP	D	4	BUN	AL	mg/dL	1.6	0.04	*
[68]	1	Broiler	1200	CP	D	16	BUN	AL	U/L	1.3	0.50	*
[69]	1	Broiler	30	CP/Gly	D	6	SUN	AL	mg/dL	3.4	-	*
[70]	1	Broiler	960	Lys	D	2	SUN	AL	mmol/L	1.0	0.05	*
[71]	1	Broiler	360	Lys	TI	6	SUN	AL	-	-	-	*
[65]	1	Growing Pig	102	Val	D	6	PUN	AL	mg/dL	6.0	0.57	+
[65]	1	Growing Pig	120	Ile	D	6	PUN	AL	mg/dL	6.3	0.57	+
[72]	1	Growing Pig	216	CP	D	3	PUN	R	mg/dL	20	1.50	*
[73]	1	Growing Pig	360	Lys	TI	5	PUN	R	mg/dL	30	3.00	*
[73]	2	Growing Pig	360	Thr	TI	5	PUN	R	mg/dL	20	3.00	*
[63]	1	Growing Pig	17	CP	D	1	BUN	R	mg/100 mL	40	-	*
[74]	1	Growing Pig	12	Lys	D	2	PUN	R	mg/100 mL	12	1.00	*
[74]	2	Growing Pig	12	Lys	D	4	PUN	AL/R	mg/100 mL	20	3.50	*
[66]	1	Growing Pig	120	Lys	D	5	SUN	AL	mg/dL	20	0.74	*
[66]	1	Growing Pig	120	Lys	D	5	SUN	AL	mg/dL	20	0.74	*
[66]	2	Growing Pig	21	CP	D	21	SUN	AL	mg/dL	22	1.96	*
[66]	2	Growing Pig	21	CP	D	1	SUN	AL	mg/dL	22	1.96	*
[75]	1	Growing Pig	180	Trp	TI	6	PUN	R	mmol/L	1.0	0.10	*
[75]	2	Growing Pig	120	Trp	TI	6	PUN	R	mmol/L	3.0	0.25	*
[75]	3	Growing Pig	144	Trp	TI	6	PUN	R	mmol/L	2.8	0.11	*
[76]	1	Growing Pig	150	Lys	TI	6	PUN	AL	mmol/L	6.6	0.43	*
[76]	2	Growing Pig	150	sAA	TI	5	PUN	AL	mmol/L	2.8	0.12	
[76]	3	Growing Pig	168	sAA	TI	5	PUN	AL	mmol/L	2.8	0.12	*
[77]	1	Growing Pig	12	Lys	D	-	SUN	AL	-	-	-	*
[77]	1	Growing Pig	360	Lys/Trp/Thr	TI	4	SUN	AL	-	-	-	*
[78]	1	Growing Pig	60	Lys	D	6	PUN	AL	mg/100 mL	20	1.60	*
[69]	1	Growing Rabbits	56	Lys	TI	2	PUN	AL/R	mg/dL	15	0.45	*
[69]	1	Growing Rabbits	56	Lys	TI	2	PUN	AL/R	mg/dL	15	0.45	*
[79]	1	Hens	720	Val	D	5	BUN	AL	Mm	5.0	1.10	*
[80]	1	Hens	225	Arg	TI	5	PUN	AL	mmol/L	10	0.80	*
[81]	1	Lactating Sows	72	Val	D	6	PUN	R	mg/L	280	83.0	+
[82]	1	Lactating Sows	163	sAA/Lys	D	5	PUN	R	Mm	3.8	018	
[64]	1	Lactating Sows	24	Lys	D	-	PUN	AL	-	-	-	*
[56]	1	Lactating Sows	72	Lys	D	6	PUN	AL	mg/dL	7.5	1.20	*
[56]	2	Lactating Sows	12	Lys	D	3	PUN	AL	mg/dL	10	0.48	*
[83]	1	Lactating Sows	4	Lys	D	6	BUN	AL/R	mmol/L	5	0.20	*
[83]	1	Lactating Sows	4	Lys	D	6	BUN	R	mmol/L	5	0.20	*
[83]	2	Lactating Sows	4	sAA	D	6	PUN/BUN	AL				*
[84]	1	Lactating Sows	8	Thr/Val	TI	4	PUN	AL	mg/dL	6.5	0.30	*
[84]	2	Lactating Sows	12	Lys/sAA/Gly	TI	3	PUN	AL	mg/dl	6.0	0.30	*
[84]	3	Lactating Sows	12	sAA/Gly	TI	2	PUN	AL	mg/dl	8.0	0.30	*
[85]	1	Lactating Sows	12	Lys	D	-	PUN	AL	-	-	-	*
[86]	1	Pregnant Sows	15	Lys/Thr	D	4	PUN	R	mg/100 mL	9.0	-	*

CP: crude protein; Gly: Glycine; Lys: Lysine; Val: Valine; Ile: Isoleucine; Thr: Threonine; Val: Valine; sAA: Sulphur amino acids; D: dietary; TI: True ileal; UN: Urea nitrogen; PUN: Plasmatic urea nitrogen; BUN: Blood urea nitrogen; SUN: Serum urea nitrogen; AL: Ad libitum; R: Restricted; X: mean; SE: Standard error. * Effects was significant *p* < 0.05; + Tendency was observed.

## Data Availability

Not applicable.

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
