# Peer review of "Urea Nitrogen Metabolite Can Contribute to Implementing the Ideal Protein Concept in Monogastric Animals"

_animals, 2022, doi:10.3390/ani12182344_

Round 1

Reviewer 1 Report

Dear Author, 

your manuscript is interesting and, in my opinion, only few changes are required before publication.

Please, take into account my comments.

L. 31 define PCC

L 61 add "In addition, there is an increasing interest in the use of low-protein diets as a tool to improve the sustainability of animal production (Attia et al., 2020, doi 10.3390/ani10060973 

L 85 add a dot after (McDonald et al.). In addition, I'm not sure this is the right format of the cited literature

L 97 check cited literature

L 128 McDonald et al. check

L 137 change "," with ";"

L 142-143 change the period "females produce higher milk production"

L 161 add a space "animal(e.g....)

L 185 amino acids instead of acids

L 186 change "between amino acids" as "among them"

L 192 change ";" with ","

L 243 check the literature format

L 258 change "de" as "of"

L 261 check the literature format also along the MS

L 297 "to the requirements"

L 298 "Other trials"

L 299 "and the reproductive"

L 304 close the parenthesis

Add some lines about your statistical approach.

Author Response

Reviewer 1:

Dear Author, 

your manuscript is interesting and, in my opinion, only few changes are required before publication.

Please, take into account my comments.

Dear Reviewer 1, we would like to thank you for all the effort and dedication in reviewing our manuscript. We would like to inform you that we have considered all your comments and have incorporated them into this review, with the aim of improving it

  1. 31 define PCC. Done.

L 61 add "In addition, there is an increasing interest in the use of low-protein diets as a tool to improve the sustainability of animal production (Attia et al., 2020, doi 10.3390/ani10060973 Done.

L 85 add a dot after (McDonald et al.). In addition, I'm not sure this is the right format of the cited literature Done.

L 97 check cited literature Done.

L 128 McDonald et al. check Done.

L 137 change "," with ";" Done.

L 142-143 change the period "females produce higher milk production" Done.

L 161 add a space "animal(e.g....) Done.

L 185 amino acids instead of acids Done.

L 186 change "between amino acids" as "among them" Done.

L 192 change ";" with "," Done.

L 243 check the literature format Done.

L 258 change "de" as "of" Done.

L 261 check the literature format also along the MS Done.

L 297 "to the requirements" Done.

L 298 "Other trials" Done.

L 299 "and the reproductive" Done.

L 304 close the parenthesis Done.

Add some lines about your statistical approach. Done.

We hope that these modifications have improved the manuscript

Once again, thank you for your effort and dedication.

The authors.

Reviewer 2 Report

Dear authors,

The presented subject is interesting and the article is very well structured. Being a review article, the experimental methodology does not need to be evaluated. I believe that the article is prepared correctly, the bibliographic resources are adequate, the study is well argued, and the Conclusions follow those mentioned in the article, although for this section, the expression could be improved.

I noticed that recommendations were also issued for a possible standardization of the method. I think this is desirable, but there is a long way to reach this point. Nevertheless, I congratulate you on the initiative.

Author Response

Reviewer 2:

The presented subject is interesting and the article is very well structured. Being a review article, the experimental methodology does not need to be evaluated. I believe that the article is prepared correctly, the bibliographic resources are adequate, the study is well argued, and the Conclusions follow those mentioned in the article, although for this section, the expression could be improved.

I noticed that recommendations were also issued for a possible standardization of the method. I think this is desirable, but there is a long way to reach this point. Nevertheless, I congratulate you on the initiative.

Thank you very much for your comments. It is a pleasure that the work is accepted by the scientific community. Thanks for everything.

Reviewer 3 Report

This work aims to critically analyze how urea nitrogen metabolite can contribute to accurately implement the ideal protein concept in pigs, poultry, and rabbit nutrition.

While a relative good scientific study is presented, the originality of the study is questioned as it does not seem to provide novel findings.

To improve the quality of the work, I suggest you also talk a little about urea nitrogen (ideal protein concept) and the feeding of monogastric animals during heat stress in the context of climate change that has become a reality. It is known that the metabolism of amino acids in excess produces the most heat among all nutrients, negatively influencing feed intake, productivity, health, well-being, immunity and the quality of animal production. A good balance of feed in amino acids (ideal protein) would reduce the heat of metabolism, the animals withstanding thermal stress better.

The conclusions of the conducted research are sufficiently clear.

Please review the References chapter. I believe that many bibliographic titles are old and irrelevant: 1952, 1956, 1959, 1966, 1971, 1972, 1974, 1977, 1978 …….. (Mertz, E. T., Beeson, W. M. & Jackson, H. D. Classification of essential amino acids for the weanling pig. Archives  of Biochemistry and Biophysics 38, 121–128 (1952); Johnson, D. & Fisher, H. The Amino Acid Requirement of the Laying Hen. The Journal of Nutrition 60, 275–282 (1956)).

Author Response

Reviewer 3:

Dear Reviewer 3, we would like to thank you for all the effort and dedication in reviewing our manuscript. We would like to inform you that we have considered all your comments and have incorporated them into this review, with the aim of improving it

This work aims to critically analyze how urea nitrogen metabolite can contribute to accurately implement the ideal protein concept in pigs, poultry, and rabbit nutrition.

While a relative good scientific study is presented, the originality of the study is questioned as it does not seem to provide novel findings. To improve the quality of the work, I suggest you also talk a little about urea nitrogen (ideal protein concept) and the feeding of monogastric animals during heat stress in the context of climate change that has become a reality. It is known that the metabolism of amino acids in excess produces the most heat among all nutrients, negatively influencing feed intake, productivity, health, well-being, immunity and the quality of animal production. A good balance of feed in amino acids (ideal protein) would reduce the heat of metabolism, the animals withstanding thermal stress better. The conclusions of the conducted research are sufficiently clear. Done. We have added a paragraph in the Perspectives (L407). Thank you very much for your support.

Please review the References chapter. I believe that many bibliographic titles are old and irrelevant: 1952, 1956, 1959, 1966, 1971, 1972, 1974, 1977, 1978 …….. (Mertz, E. T., Beeson, W. M. & Jackson, H. D. Classification of essential amino acids for the weanling pig. Archives  of Biochemistry and Biophysics 38, 121–128 (1952); Johnson, D. & Fisher, H. The Amino Acid Requirement of the Laying Hen. The Journal of Nutrition 60, 275–282 (1956)). Done. We have reviewed the bibliography and eliminated the one considered as not relevant

L87-94: I believe that these lines can be eliminated – it is too much for a current scientific paper. Done. The paragraph has been modified under yours recommendations.

L117-157: The discussion is taken to a much too elementary level – course for students. Done. This part of the discussion has been reconsidered. The paragraphs have been modified under yours recommendations.

We hope that these modifications have improved the manuscript

Once again, thank you for your effort and dedication.

The authors.

Round 2

Reviewer 3 Report

The manuscript has been sufficiently improved to warrant publication in Animals.